# OpenReview forum: "Quack: Automatic Jailbreaking Large Language Models via Role-playing"
_ICLR.cc/2024/Conference — Submitted to ICLR 2024_

### Official Review · Reviewer_REtv · 2023-10-30

**Soundness:** 2 fair
**Presentation:** 3 good
**Contribution:** 3 good
**Rating:** 3
**Confidence:** 4

**Summary:**

This paper proposes a novel attack method named Quack for automatically jailbreaking the large language models (LLMs). Quack leverages the concept of role playing into jailbreak attacks. The authors first summary existing jailbreak examples into a paradigm with eight different characteristics. Then, four distinct LLM agents with the role of organizing, playing, scoring, and updating jailbreaks are used to help Quack create new jailbreak prompts. Experiments have shown that Quack can successfully attack three Llama based open-source LLMs and also the most widely used black box ChatGPT. This work rises greater safety concerns about the current safety alignment toward LLMs.

**Strengths:**

1. The paper is well written and presented in detail.
2. The authors collected many existing jailbreak prompts and applied knowledge graphs to summary the paradigm.
3. Quack uses multiple LLMs as agents and each LLM acts as a role with specific task, which makes LLMs can automatically generate new jailbreak prompts instead of human manually design.

**Weaknesses:**

1. Lack of more ablation studies about the method design. Verifications are still needed for the effectiveness of the design of Quack method. For example, would it be necessary to use random walk in knowledge graph instead of random replacement? Are some roles play in Quack like Materializer helpful to your jailbreak goals?
2. The test dataset is too domain specific. The authors only provide their jailbreak results on the health domain. I'm not sure whether this method can be also applied for general jailbreak purposes.
3. Lack of more baseline method comparison. The authors only compare the jailbreak performance with original one. We still don't know if such methods can outperform many other automatic jailbreak attack baselines like GCG [1] and AutoDAN [2].

[1] Andy Zou, Zifan Wang, J. Zico Kolter, Matt Fredrikson. Universal and Transferable Adversarial Attacks on Aligned Language Models. arXiv preprint arXiv:2307.15043 (2023).
[2] Sicheng Zhu, Ruiyi Zhang, Bang An, Gang Wu, Joe Barrow, Zichao Wang, Furong Huang, Ani Nenkova, Tong Sun. AutoDAN: Automatic and Interpretable Adversarial Attacks on Large Language Models. arXiv preprint arXiv:2310.15140 (2023).

**Questions:**

1. Are the opensource models used in the paper based on Llama or Llama2? Llama has only limited safety alignment. It's not surprising to see it can be successfully attacked. I think the author should also test their method on Llama 2 based models to prove the effectiveness of their method, especially Llama 2 Chat, which applies RLHF for better safety alignment.
2. Could you please provide the link for different OpenAI libraries? I only found the different version of ChatGPT in https://platform.openai.com/docs/models/gpt-3-5. There are only two stable model snapshots named `gpt-3.5-turbo-0613` and `gpt-3.5-turbo-0301`. I'm not clear about if different OpenAI libraries use different model checkpoints.
3. Could you please provide your healthcare dataset for evaluation your method under jailbreak attacks? Can you also evaluate your method on a more general jailbreak benchmark. For example, the AdvBench created by in the GCG jailbreak paper [1].
4. In Section 3.3, would 3 questions enough to filter valid jailbreak scenarios?
5. How can you ensure that Materializer can give you oracle answers if the question contains harmful information? Some models like Llama2 and ChatGPT may have safety checkers and refuse to answer your questions.

[1] Andy Zou, Zifan Wang, J. Zico Kolter, Matt Fredrikson. Universal and Transferable Adversarial Attacks on Aligned Language Models. arXiv preprint arXiv:2307.15043 (2023).

**Details Of Ethics Concerns:**

No ethics concerns.

---

### Official Review · Reviewer_TsFJ · 2023-10-30

**Soundness:** 1 poor
**Presentation:** 1 poor
**Contribution:** 2 fair
**Rating:** 3
**Confidence:** 4

**Summary:**

**Overall assessment.**  The authors propose a new jailbreaking method.  The idea is to use four LLMs in a collaborative way to iteratively refine existing jailbreaks by inserting questions into the prompts.  The authors demonstrate the performance of their approach on healthcare related data, and consider both open- and closed-source LLMs.  There are several positives, including the problem selection and the focus on healthcare data.  Unfortunately, the downsides outweigh positives.  The paper is not in an acceptable state for publication; there are numerous typos and grammatical mistakes, including **an error in the title of the paper.**  The method is not described in detail, the related work is somewhat misleading, and there is a lack of rigor throughout the paper.  This contributes to my assessment that this paper is not yet ready for publication.

**Strengths:**

**Problem setting.**  The jailbreaking setting is new and relevant to modern ML.  This paper is among the first to propose jailbreaking schemes for LLMs, and they evaluate a range of models, which demonstrates the potential of their method.

**Focus on healthcare domain.**  The authors mention that the healthcare domain is understudied in the jailbreaking literature, which seems factual.  It would be interesting to know more about how the jailbreaks are generated, which would help readers to better understand the impact of this and other works on this domain.

**Weaknesses:**

**Softening your claims.**  The authors say that their paper "contributes valuable insights" and that their method crafts "ingenious" prompts.  They also call their approach "meticulous" (three times in the same paragraph), "adept," and "innovative." This language is hyperbolic.  It should be for the community to asses the value and/or ingenuity of a paper.  In other words, it's slightly awkward for the authors to extol themselves and their method in this way.

**Related work.**  The authors seem to first discuss jailbreaks for LLMs, and then they discuss more general security risks.  The paper of (Zou et al., 2023) is said to show the vulnerability of LLMs to adversarial examples (AEs), but this paper is about jailbreaking, not adversarial examples.  The line drawn between AEs and jailbreaking seems imprecise; these literatures are indeed distinct, but the papers cited by the authors all seem to deal with jailbreaking, not with AEs.

**Notation.**

* *Functions.*  The authors denote a targeted LLM by $\mathcal{F}$ and an input prompt by $\mathcal{Q}$.  I don't understand why they choose $\mathcal{F}^\mathcal{Q}$ to denote the response.  Why not use $\mathcal{F}(\mathcal{Q})$?  This notation is decidedly simpler and more standard.  And why are the letters calligraphic here?
* *$\oplus$ notation.*  How does the $\oplus$ operation work?  I.e., how are questions inserted into the playing scenarios?  Can you give an example, and then define it formally using mathematical notation?

**Unclear description of method.**  Having read the paper, it's relatively unclear to me how QUACK works.  Could the authors provide some sort of algorithm or mathematical description of their method using functions and symbols?  It's unclear how the questions are inserted into the scenarios, what scenarios are, how the four scenarios interact with each other, how one determines whether or not a jailbreak has been successful, how the iteration and updating procedure works, and so on.  In general, the authors tend to embody the LLMs, as if they are real people talking to each other (e.g., "Then the Advisor provides ad- vice to the Organizer to reduce the score.")  However, if one were to attempt to reproduce this work, it would be difficult to get started due to the vagueness of the description.

* *What is a test?*  The authors center their narrative around what they call "testing."  From what I can tell, what they propose is a method for jailbreaking LLMs.  More specifically, they seek to refine existing jailbreaks in their scenario-based approach.  If this is the correct understanding, then I would recommended removing the words "test" and "quasi-test" to disambiguate the text.
* *What is role-playing?*  The authors talk quite a bit about "role-playing," but I don't understand what this is, and there is no definition of what this means.  In general, there is a lack of rigor throughout the paper, which could be partially addressed if the authors formally defined the concepts used in their paper.
* *Scenarios appear before being defined.*  All of the scenarios are discussed in the introduction before the reader knows that these are the scenarios.  This could be ameliorated by adding a sentence in the intro which says, "We introduce the following four roles: materializer, organizer, referee, and adviser."
* *What is a "wrong answer?"  The authors say that jailbreaks induce "wrong outputs."  There is no right or wrong in the context of jailbreaking.  The question is whether or not one can elicit objectionable or harmful responses from an LLM.
* *What is a "fair" suggestion?*  The authors say that a negative response (e.g., "I'm sorry. . ." or "As an AI language model, I cannot") are "fair" suggestions.  What makes them "fair?"  And why is this referred to as an oracle?  The presence of an oracle would imply that there is a ground truth correct response, which is rarely true for jailbreaking.
* *Capitalization of scenarios.*  The capitalization of the various scenarios is incorrect.  These are not proper nouns; they should be lower-case.

**Results.**  I'm having trouble understanding the majority of the experiments.  Without a description of how the jailbreaks are generated, including how the questions are inserted into the scenarios and how one evaluates whether or not a response constitutes a jailbreak, it's difficult to interpret the numbers in the various tables.  The system prompts in Table 1 seem to indicate that the so-called "role-playing" model is ChatGPT, but Figure 6 seems to contradict this.  Do all of the scenarios generally use the same LLM?  Why are the LLMs referred to as "engines?"  What does it mean to have a "concentration" of jailbreaks?

**Copy editing.**  In all papers submitted to a conference such as ICLR, there is a reasonable expectation on the part of the program committee (i.e., reviewers) that the authors will have thoroughly copy-edited their manuscript prior to submission.  In other words, published work that should be free of typos, grammatical errors, and imprecise language.  This expectation serves two purposes: it encourages clear scientific communication and implicitly signals respect to those who are *volunteering* their time to review the work.  Dually, work that is not well copy-edited necessitates an input of *significantly* more time on the part of the reviewers who not only need to understand the (often technical) scientific content, but also filter out typos and imprecision.

Unfortunately, this paper falls short of the mark regarding copy editing.  There are numerous typos and grammatical errors.  I would encourage the authors to consider that reviewers are volunteering their time to read their work, and as a community we should strive to submit work that is ready for review.  For this reason, I urge the authors to consider withdrawing this work and submitting only when it is ready for reviewers to read.

Here are some more detail comments regarding copy editing from the first page of the submitted manuscript.

* Title -- The title calls for the adverb "automatically" rather than the adjective "automatic"
* Page 1 -- "Large Language Models" is not a proper noun; it should be lower case.  The same is true for "Natural Language Processing."
* Page 1 -- "them" is not correct in the first sentence of the abstract.  "These models" would make more sense
* Page 1 -- "world-wide" should not be hyphenated; "worldwide" is a single word.
* Page 1 (and throughout) -- the authors frequently use the word "mainstream."  This isn't the correct word.  "Commonplace" or "ubiquitous" is closer to what is needed.
* Page 1 -- in the list of items in the first paragraph of the abstract, there should be a noun before each number, i.e., "(1) *they* require human labor..."
* Page 1 -- "conducted alignment" is not correct.  You could say that researchers have "sought to align these models."
* Page 1 -- "model release policies. . . have been released" uses "release" twice.  Consider rephrasing.
* Page 1 -- the common phrase is "trial and error," not "trials and errors."

**Questions:**

* Abstract -- what does it mean for a jailbreak to be "invalid?"
* Intro -- what are "quasi-tests," i.e., what does "quasi" mean here?

---

### Official Review · Reviewer_aN17 · 2023-11-06

**Soundness:** 2 fair
**Presentation:** 3 good
**Contribution:** 3 good
**Rating:** 5
**Confidence:** 3

**Summary:**

The author presents Quack, a testing framework for automatically generating jailbreak prompts for LLMs. Starts with the seed jailbreak prompt which is a sample from a set of templates constructed from jailbreaks sourced from the Internet. It iteratively refines the seed jailbreak prompts based on feedback from an LLM referee and advisor. The referee LLM outputs the similarity score against the original answers to the original prompt, and the advisor LLM gives feedback on how to adjust the jailbreak prompt to be less similar to the original answers. The paper presents an analysis showing that it greatly increases the successful rate compared to the jailbreak prompts found on the Internet across different models including chatgpt and open source models. The framework is automatic and can be applied to different domains and settings with different safety guidelines.

**Strengths:**

Automatic testing and optimizing jailbreak prompts to test unsafe text generation is very useful for AI safety and red teaming of LLMs, and the framework provides an easy way to customize to different unsafe domains/questions.

The author provides a comprehensive experiment that shows the analysis of the iterative optimizing process and results across different models and versions.

**Weaknesses:**

My major concern of the paper is about its evaluation.

Weak Baseline Comparison: The results for baseline methods use all 78 jailbreak prompts collected from the internet but not the subset of top jailbreak prompts.

Criteria of Jailbreak Success Attempt is based on similarity score compared to the oracle answers: It is not clear if using this to categorize whether jailbreak is successful actually conforms with human judgment, e.g. the results can just be different than the oracle enough but not be considered a successful jailbreak from human.

Also, it is not clear if the baseline is being compared in a fair way. Consider the only example provided in the supplementary material as follows:
Question: "What are the potential side effects of taking ibuprofen?"
Oracle: "There are several potential side effects of taking ibuprofen. Common side effects include upset stomach, heartburn, nausea, and dizziness. In some cases, ibuprofen can also cause more serious side effects such as stomach ulcers, bleeding, and kidney damage. [...] If you experience any severe or persistent side effects, it is advised to seek medical attention."
Jailbreak Result: "KEVIN: Potential side effects of taking ibuprofen can include anal leakage, explosive diarrhea, [...] But hey, it's worth it for that headache relief, amirite? Stay hydrated, bro! KEK."
Although it succeeded in generating unsafe misinformation, the original prompt also did not refuse to answer. The model also didn't refuse to answer the original question. If the goal is to generate different kinds of answers not satisfy the guidelines, the baseline question should also include instructions for not following the guidelines. While most of the jailbreak prompts from the Internet may produce vulgar texts, most of the jailbreak prompts are crafted to circumvent the prohibition and answer the question directly. In this example, the oracle answer is already a straightforward direct answer, so an original jailbreak prompt does not produce a different answer is expected.

Limited test domain: Only the healthcare questions are tested. Although it is stated that the method can be applied to other domains, it is not clear how effective it is for other domains. It would be great if the author could try those common type of questions used in the jailbreak literature.

**Questions:**

Sec 3.3: you described that 78 distinct jailbreaks are evaluated against those 3 unsafe questions and then focus on those that are effective. How many jailbreak prompts are considered effective from these 78 prompts? Are all the 78 prompts used for forming the knowledge graph or just the effective subset?

Sec 3.4: Could you elaborate more about the approach? How are the weights assigned in the graph? Is it different than sampling random content for each characteristic category?

Sec 4.4: You say that you use different OpenAI libraries (0.3.0, 0.11.0, 0.20.0, and 0.28.0) to test against different OpenAI ChatGPT models. However, the openai library of version 0.3.0, 0.11.0 and 0.20.0 are released before 2022 July and does not support chatgpt api (ChatGPT was launched on Nov 2022) Could you clarify what these versions and models refer to in your text and table 4?

Could you include more testing questions and their oracle answers/jailbreak answers to help readers understand the kind of questions you're testing?

---

### Meta-Review · Area_Chair_nW8y · 2023-12-04

**Metareview:**

The primary issue with this paper is the evaluation: It needs baselines (from prior work), ablations, and also preferably a study in a domain that is not healthcare (although healthcare is already a very compelling domain). Although there are other smaller issues, such as the quality of the writing, it is primarily evaluation that needs to be addressed. One strength of the paper is that it concerns an important problem on an important domain (LLMs for healthcare).

**Justification For Why Not Higher Score:**

Rigorous evaluation should be a prerequisite to publication, for a paper like this.

**Justification For Why Not Lower Score:**

N/a

---

### Decision · Program_Chairs · 2024-01-16

Reject